



# Using isotopes to understand evaporation, moisture stress and re-wetting in catchment forest and grassland soils of the summer drought of 2018.

Lukas Kleine[1,2], Doerthe Tetzlaff[1,2], Aaron Smith[2], Hailong Wang[3], and Chris Soulsby[4,2]

[1]Department of Geography, Humboldt University of Berlin, Rudower Chaussee 16, 12489 Berlin, Germany
[2]Department of Ecohydrology, Leibniz Institute of Freshwater Ecology and Inland Fisheries, Müggelseedamm 310, 12587 Berlin, Germany
[3]School of Civil Engineering, Sun Yat-sen University, 135 Xin'gang Xi Road, Guangzhou, 510275, China
[4]Northern Rivers Institute, University of Aberdeen, St. Mary's Building, Kings College, Old Aberdeen, AB24 3UE, Scotland

**Correspondence:** Lukas Kleine (l.kleine@igb-berlin.de)

**Abstract.** In drought sensitive lowland catchments, ecohydrological feedbacks to climatic anomalies can give valuable insights into ecosystem functioning in the context of alarming climate change projections. However, the dynamic influences of vegetation on spatio-temporal processes in water cycling in the critical zone of catchments are not yet fully understood. We used stable isotopes to investigate the impacts of the 2018 drought on dominant soil-vegetation units of the mixed land-use

Demnitzer Mill Creek (DMC, NE Germany) catchment (66 km$^2$). The isotope sampling was carried out in conjunction with hydroclimatic, soil, groundwater, and vegetation monitoring. Drying soils, falling groundwater levels, cessation of stream flow and reduced crop yields demonstrated the failure of catchment water storage to support "blue" and "green" water fluxes. We further conducted monthly bulk soil water isotope sampling to assess the spatio-temporal dynamics of water soil storage under forest and grassland vegetation. Forest soils were drier than the grassland mainly due to higher interception and transpiration

losses. However, the forest soils also had more freely draining shallow layers, and were dominated by rapid young (age < 2 months) water fluxes after rainfall events. The grasslands soils were more retentive and dominated by older water (age > 2 months), though the lack of deep percolation produced water ages ∼1 year under forest. We found the displacement of any "drought signal" within the soil profile limited to the isotopic signatures and no displacement or "memory effect" in d-excess over the monthly time step, indicating rapid mixing of new rainfall. Our findings suggest that contrasting soil-vegetation as-

semblages communities have distinct impacts on ecohydrological partitioning and water ages in the sub surface. Such insights will be invaluable for developing sustainable land management strategies appropriate to water availability and build resilience to climate change.



## 1 Introduction

Climate change provides an urgent impetus for an improved understanding of ecohydrological interactions in areas where water is limited (Wang et al., 2012). Increasing temperatures and reduced rainfall in the growing season are affecting extensive regions (Tetzlaff et al., 2013); in some cases, causing natural vegetation communities to adapt by changing their composition, distribution and physiology (Wookey et al., 2009). Where vegetation is managed for forestry and agriculture, land use strategies may need to adapt to build resilience towards newly evolving climate regimes. This includes choice of species, crop rotation

cycles and sustainable production targets (Stoate et al., 2009). As well as constraining biomass productivity, such strategies will also have implications for the residual water available to maintain groundwater recharge, river flows and associated ecosystem services. In summer 2018, an exceptional drought over much of Europe set new, long-term meteorological records causing significant loss of agricultural production, water shortages and low river flows over extensive areas (Imbery et al., 2018). This drought previewed the warmer, drier conditions that climate change is expected to deliver across much of Central Europe as

the 21$^{st}$ Century progresses. For a future where water resources might become less reliable, conceptualisation of the dynamic interactions between vegetation, soils and water fluxes from stores in various ecosystem compartments needs to be improved and is a major focus of current "critical zone" science (Penna et al., 2018). Understanding local environmental factors, like how atmospheric water demand is modulated by vegetation cover, is a prerequisite to better managing the effects of droughts (Mishra and Singh, 2010). Stable isotopes in the water molecule have been successfully used to trace water fluxes in the soil-

plant-atmosphere-continuum (Sprenger et al., 2017), and can reveal important process insights into ecosystem water cycling (Dubbert and Werner, 2019). Stable isotopes of hydrogen and oxygen are often seen as ideal tracers as they are an integral part of the water molecule itself. Isotopes are conservative tracers and not altered by chemical reactions, but only by mixing and fractionation. Numerous studies have applied stable water isotopes to constrain water sources and fluxes in the unsaturated zone. Complementary to hydroclimatic monitoring, stable isotopes as environmental tracers can provide insights into ecohy-

drological processes in the "critical zone" (Grant and Dietrich, 2017). They have been used to investigate evaporation (Allison and Barnes, 1983; Barnes and Allison, 1988), groundwater recharge (Koeniger et al., 2016), weathering influence on flow paths (Bullen and Kendall, 1998) as well as water ages (Tetzlaff et al., 2014; Sprenger et al., 2019b), plant water uptake (Rothfuss and Javaux, 2017) and the partitioning of evapotranspiration (Kool et al., 2014; Xiao et al., 2018). Highly seasonal dynamics of soil water ages and their dependency on soil water storage have further been investigated via stable isotope modelling (Sprenger

et al., 2018). Isotopes were also used to examine the influence of vegetation on soil bulk (Oerter and Bowen, 2019; Sprenger et al., 2017) or other components like throughfall (Soulsby et al., 2017) often supplemented with xylem water isotope data (Geris et al., 2015; Brooks et al., 2010; Goldsmith et al., 2019). The opportunity to measure stable isotope ratios via inexpensive laser absorption spectroscopy has facilitated these new applications. However, there are still unresolved problems related to sampling (Sprenger et al., 2015) or extraction (Orlowski et al., 2016, 2018) of water from complex matrices like soil or plant

tissue. Laboratory routines for the direct-equilibrium method (Wassenaar et al., 2008) use the state of isotopic equilibrium between liquid and gaseous water in a closed system to determine the isotopic signature of the liquid soil water. They have been developed (e.g., Hendry et al., 2015) and applied successfully in several studies (e.g., Klaus et al., 2013; Sprenger et al., 2017;





Stumpp and Hendry, 2012). On-going efforts are being made to solve various problems associated with the different methods

for soil water isotope analysis, and results have to be interpreted accordingly (Gaj and McDonnell, 2019; Gralher et al., 2018;

Sprenger et al., 2015). This study focuses on the long-term monitoring site Demnitzer Millcreek catchment (DMC), a mixed

land use catchment located south-east of Berlin in Brandenburg, Germany. As the exceptional drought developed in summer

2018, we monitored moisture dynamics in drying soil profiles under different land cover types, falling groundwater levels and

decreasing stream flows. Crucially, we used stable isotopes from different waters in the latter stages of the drought to address

the specific objectives of this study:

1. To assess the development and progress of the drought and the subsequent recovery on soil water storage.

2. Explore, using bulk soil water isotopes, the evolution of the evaporation signal of the drought and its "memory" effect fol-

lowing infiltration & mixing with new precipitation during re-wetting.

3. Discuss the implications of ecohydrological processes for the response times and recovery of hydrological stores in the DMC

catchment by combined use of hydroclimatic and isotope data.

## 65  2   Study site

Our study was based in the 66 km$^2$ Demnitzer Millcreek catchment (DMC) in Brandenburg, north east Germany (52°23'N

14°15'E), 55 km south-east of Berlin. This long-term study site is a tributary of the River Spree and one of the few headwaters

in the region that does not originate in a lake but in a network of agricultural drainage channels. Catchment orientation is NNE

– SSW with elevation from 38 to 83 m above sea level and a low average slope of less than 2 %.

Located in the Northern European Plain, the geology of the catchment is strongly influenced by the Pleistocene glaciation.

The catchment outlet is situated in the Berlin glacio-fluvial valley near Berkenbrück, where the DMC surface runoff drains

into a small lake (Dehmsee) and subsequently into the River Spree. The geology of the upper catchment is dominated by

unconsolidated sediments of ground moraine material. Important factors for nutrient cycling in this landscape are kettle hole

lakes (Nitzsche et al., 2017) and wetlands (Smith et al., 2020). The stream network is embedded in fluvial and periglacial

deposits surrounded by basal tills with intermittent riparian peat fens in valley bottom areas. The northern catchment is mainly

characterised by eutric soils and silty brown earths. Next to the stream, sandy gleysols or peaty histosols are dominant. DMC has

a seasonal and strongly continental climate, with cold winters (mean air temperatures in January and July are 0.2 ° and 19 °C,

respectively). Precipitation is dominated by convective summer events and low intensity winter rain, with generally less than

10 % of the annual total occurring as snowfall. Potential evapotranspiration (PET) commonly exceeds average precipitation

and runoff coefficients are typically < 10 % of annual precipitation (Smith et al., 2020). Non-irrigated arable land (mainly

winter cereals, maize) dominates the upper catchment and contributes 58 % of the area. Further downstream, the cover of

mixed coniferous and deciduous forests increases. The stream traverses several peat fens that were used as pasture. Manmade

connections of disconnected glacial hollows to the stream network altered the total channel length from 20 km in 1780 to 88 km

at the present day (Nützmann et al., 2011) to supply mills and gain new arable land by draining. The catchment has been subject

of various studies investigating e.g. CO$_2$ saturation (Gelbrecht et al., 1998), influences of wastewater treatment (Gücker et al.,





2014), other historical anthropogenic impacts (Nützmann et al., 2011) and the impact of beaver re-colonisation (Smith et al., 2020). This study focuses on two plots with contrasting landcover in close spatial proximity to each other (∼400 m) and the stream (Figure 1). The two experimental plots are forested (FA) and covered by grassland (GS). FA is dominated by mature oak trees (Quercus robur), and includes other tree species such as Scots Pine (Pinus sylvestris) and red oak (Quercus rubra).

GS is pasture that is harvested once a year. Distance to stream differs between GS (∼15 m) and FA (∼90 m). GS has eutric arenosol (humic, transportic) soil whereas the FA soil is a lamellic brunic arenosol (humic) according to the World Reference Base (WRB) classification. GS is characterized by higher clay contents in the upper soil, a higher pH, and narrower C/N ratio than FA. There is also a shift in pH at FA due to the presence of calcite at the lowest layer.

## 3   Data and methods

An automatic weather station AWS (Environmental Measurement Limited, UK), located in the NW of the catchment, was used to record meteorological data (e.g. net radiation, air temperature, precipitation, ground heat flux, relative humidity) every 15 minutes. To monitor transpiration rates from trees at FA, a sap-flow measuring system with 32 sets of Granier-type (Granier, 1987) sensors (Thermal Dissipation Probes, Dynamax Inc., Huston, USA) was installed in 13 trees during the growing season from 21.4.18 to 23.10.18. Sensors were installed at approximately 1.3 meter above ground. The tree diameter was also

measured at this height (DBH; mean: 76 cm; SD: 35 cm). All sensors consisted of two thermometers installed in the sapwood in 4 cm vertical distance from each other and were shielded from external sources of temperature change (e.g. radiation). The upper thermometer was heated and differences in temperature were collected hourly with a CR1000 data logger (Campbell Scientific, USA). The difference in temperature was used to calculate flux velocity and combined with the sapwood area to calculate a flux rate. Conditions of zero transpiration were determined from daily maximum temperature differences. The

resulting flux rate per unit sapwood area was adjusted to the plot using a ratio of sapwood area to forest area that was established with ten trees. Data from the AWS were used to estimate potential evapotranspiration with the FAO Penman-Monteith equation (Allen et al., 1998). To facilitate comparison, the sap-flow derived transpiration and FAO PET were normalized by subtracting the mean (T = 1.31 mm/d and PET 2.53 mm/d) and dividing the values by the standard deviation (T = 0.57 mm/d, PET 1.20 mm/d), derived from the overlapping period between 3.5.-18.10.2018. In addition, long-term monthly precipitation

data (Source: Deutscher Wetterdienst) from a nearby German Meteorological Office station (Müncheberg, 1951 – February 2019) were used to quantify drought severity and temporal development using the Standardized Precipitation Index (SPI; McKee et al., 1993). The SPI was calculated for different periods (1, 3, 6, 9 & 12 months) using a gamma distribution. Volumetric soil water content and soil temperature were measured at both sites by 72 soil moisture temperature probes (SMT-100, Umwelt-Geräte-Technik GmbH, Müncheberg, Germany) at three depths (20, 60 and 100 cm) with six replicates per site. The

probes recorded with a 15 minute frequency and a precision of ±3 % for volumetric soil water content and ±0.2 °C for soil temperature. The data was averaged and aggregated to daily values to estimate soil storage in the first meter (Figure 2) from volumetric soil moisture by weighting the upper sensors to represent 40 % of the first meter each and the lowest 20 %. One soil pit (depth > 100 cm) per plot was excavated and the profile was described following common pedological procedures.





Soil cores and composite samples were taken to determine further physical and chemical characteristics in the laboratory of
the Technical University of Berlin (Table 1). Carbon, Nitrogen and Organic Carbon concentrations from soil samples were
analysed in the Leibniz Institute of Freshwater Ecology and Inland Fisheries (IGB) laboratory. Daily samples of precipitation
for isotope analysis were collected at the AWS with a modified ISCO 3700 (Teledyne Isco, Lincoln, USA) automatic sampling
device (unshielded funnel at 1 m). Throughfall was sampled using five rain gauges (Rain gauge kit, S.Brannan & Sons, Cleator
Moor, UK), which were installed beneath the canopy at FA. Water samples in autosamplers and rain gauges were protected
from evaporation by a paraffin oil layer of a thickness > 0.5 mm (IAEA/ GNIP precipitation sampling guide V2.02 September
2014). Samples were extracted with a syringe from below the paraffin and filtered (0.2 $\mu$m, cellulose acetate) in the field and
cooled until stored at 8 °C in the laboratory. All liquid water samples were measured at the IGB laboratory with a Picarro L-
2130i cavity ring down water isotope analyser (Picarro, Inc., Santa Clara, CA, USA). To screen for interference from organics,
the ChemCorrect Software (Picarro, Inc.) was applied and contaminated samples discarded. Liquid samples were injected six
times and the first three injections discarded. The standard deviation of the three used injections per sample was on average
0.04‰ for $\delta^{18}$O and 0.14‰ for$\delta^2$H. Isotopic composition of bulk soil water was sampled at six depths (0-5 cm; 5-10 cm;
10-20 cm; 20-30 cm; 40-60 cm; 80-100 cm) on a monthly basis from October 2018 with 2 additional replicates per site. The
unequal sampling resolution across the depth profile was chosen to enable sample throughput in the laboratory while capturing
the first meter of soil and to take account of the higher heterogeneities expected in the upper soil. Initial soil bulk water sam-
pling was conducted in September 2018 using a slightly different procedure. Here, soil sampling in the forest and grassland
occurred on different dates (FA, 21.9. & GS, 7.09.2018) and with higher spatial resolution at FA of 5 cm within first 30 cm &
10 cm until 1 meter. GS had a broader resolution (10 cm until 1 m depth). Three replicates were sampled in September 2018
until a limited depth of 30 cm in the forest and 50 cm in the grassland site. Data of the September sampling was transformed
by amount weighting to be integrated in later analysis and figures. We used soil cores ($\sim$250 cm$^3$) in the topsoil (0-10 cm) to
gain sufficient water (3 ml; Hendry et al., 2015) for the lab analysis. Soil bulk water isotopes were analysed using the direct-
equilibrium (DE) routine of Wassenaar et al. (2008). Soil samples were immediately stored in sampling bags and sealed with
ziplocks instantaneously. All samples were then stored sealed and thermally isolated until being weighed, inflated with the
headspace gas and heat sealed in the laboratory. We used diffusion-tight metalized sample bags (Weber Packaging, Güglingen,
Germany) as established in other direct equilibrium studies (Sprenger et al., 2015). Synthetic dry air was utilized as inflation
atmosphere to enable a posteriori correction of biogenic gas matrix changes in the headspace (Gralher et al., 2018). After intro-
ducing dry synthetic air as headspace to the soil sample bags, the bags were heat sealed and equipped with an external silicone
septum. Samples and standards were incubated for approx. 48 hours under stable thermal conditions (21 ±1 °C) and measured
by introducing the equilibrated headspace via needle and tubing to the inlet port of the cavity ring down analyser (Picarro L-
2130i). A stable plateau in water content was ensured with quality criteria for water content standard deviation (SD) < 100 ppm,
$\delta^2$H (SD < 0.25‰), and $\delta^{18}$O (SD < 0.55‰) over 2 minutes. The real water content in the vapour was on average 27350 ppm
with a mean SD of 39 ppm and $\delta^2$H had a mean SD of 0.42‰ (for absolute corrected delta values ranging from -23.71‰ to
-91.44‰) and 0.18‰(range: -1.47‰ to 12.53‰) for $\delta^{18}$O, respectively. Groundwater levels in the catchment were monitored
4-hourly using a pressure sensor AquiLite Beaver ATP10 (AquiTronic Umweltmeßtechnik GmbH, Kirchheim/Teck, Germany)





with precision < 0.1 % and a resolution of 1 mm. The monitoring well was installed in 2000 and screened from 3.50 m to the

bottom at 5.57 m below surface. Liquid groundwater samples for isotopic analysis were obtained by monthly manual sampling

with a submersible pump (COMET-Pumpen Systemtechnik GmbH & Co. KG, Pfaffschwende, Germany). Stream water was

sampled by an ISCO 3700 at 4 pm each day (to minimise ice effects) and stored in the field covered by paraffin in the bottle until

collected and processed once a week. The stream discharge values at the Demnitz Mill were derived by using water level data

derived from pressure sensors identical in construction to the groundwater sensors and a rating curve determined by Smith et al.

(2020). Mean transit times (MTT) in the soil were estimated using stable oxygen isotope signatures of weekly precipitation

samples and monthly data of soil bulk water at different soil depths. Given the short sampling periods and monthly intervals,

the resulting MTTs are tentative, but still useful for comparing inter-site and within-profile differences. Oxygen signatures in

the soil layers were simulated from weighted precipitation inputs. Inputs were weighted by a gamma function representing the

assumed, time-invariant transit time distribution. The gamma function was fitted by maximizing the objective function using

the Kling-Gupta Efficiency statistic (KGE; Gupta et al., 2009) within predefined parameter ranges for the shape factor ($\alpha$;

0.5 to 5) and the scale parameter ($\beta$; 2 to 50); these were set to avoid MTTs shorter than the sampling frequency. The shape

of the gamma distribution enabled us to represent short- and long-term tracer input contributions (e.g., Kirchner et al., 2001)

to soil bulk water. We further excluded the first two sampling dates in the upper 10 cm of both plot soils for this analysis to

avoid implausible results due to tracer enrichment introduced by soil evaporation. We calculated young water fractions using

the fitted sine-wave method described by Von Freyberg et al. (2018), adjusting the topsoil values for the two first sampling

occasions to the precipitation input for the same reasons. The young water (Figure 2) represents the estimated fraction of water

in the sampled soil depth that fell as precipitation within the last 2-3 months.

## 4  Results

### 4.1  Hydroclimatic situation

Exceptional climatic conditions during the study period, with low precipitation and high temperatures, are reflected by the

Standardized Precipitation Index (SPI, Table 2). Values varied between "moderately wet" (1.0 to 1.49) to "extreme drought"

(-2 and less). The different SPI time windows indicate the progression of the 2018 drought in different temporal contexts and

therefore represent drought impacts on different compartments of the catchment water cycle. We found short-term monthly

SPI values ranged from -2.1 (February 2018) to 1.1 (January 2018). At this, 9 out of 14 individual months showed negative

precipitation anomalies (mean = -0.46). The heavy rainfall events in July 2018 resulted in an above average wet month (1

month, 0.9) which was also reflected in less negative 3- and 6-month SPI values. This effect did not persist, however, as the

drought index dropped again in August. The lowering was driven by the absence of significant precipitation inputs which

was reflected by negative SPI (1 month) values from August to December. Despite normal precipitation amounts, January and

February 2019 still showed marked drought characteristics over the longer-term in SPI 6, 9 & 12 (< -1). Annual patterns of

heavy convective precipitation events during summer and lower intensities during winter are also reflected in daily precipitation

amounts (Figure 2 a). The total precipitation between 21.4.2018 and 28.2.2019 was 379 mm. Precipitation occurred on 123 of





the 314 days on which the average amount was 3.1 mm. Highest daily rainfall intensities were observed in the summer with a maximum of 27 mm (7.9.2018). Mean daily air temperature during the study period ranged from 27.7 °C in summer (1.8.18) to -5.7 °C in winter (23.1.19). Daily air temperature, normalized transpiration and potential evapotranspiration show strong

seasonal patterns with maxima during summer and minima during winter. Throughout the investigated period, dynamics of normalized transpiration closely matched the normalized PET dynamics. Volumetric water content (VWC) of the FS and GS soils (Figure 2 b & c) are given as the geometric mean ($\overline{m}_g$) of accumulated daily values. The forest soil was notably drier than the grassland; and overall, the grassland soil showed much less variability and a lower drought effect on soil moisture than the forest site. The upper forest soil moisture content showed rapid responses to precipitation inputs. Further progression of

wetting fronts to depth was damped and lagged, which resulted in decreasing SD with depth (Table 3). Soil moisture in the forest at 100 cm depth showed extremely low values (min = 3.2 vol. %) and little variation during the growing season. During early December, soil moisture values here began to rise, reaching a maximum value of 10.6 vol. % in February. In contrast to the forested site, the grassland site soil moisture (Figure 2 c) was generally higher and less dynamic at all depths. The recession of soil moisture peaks at 20 cm were less steep than in the forest. GS soil moisture at 60 and 100 cm showed a steady and slow

decline in 2018 until rising in late December and beginning of January 2019, respectively. The observed groundwater level at the monitoring well (Figure 2 d) continuously declined during the drought, from -3.4 m at the beginning of the study to a minimum of -4 m in December. At the start of 2019, the groundwater table began rising, shortly after the deeper soil horizons moisture and partially recovered to a level of -3.6 m at the end of the study. Around the same time as the stabilization and rise in groundwater levels, the DMC stream began to flow again (Figure 2 d). Flows had ceased earlier in the summer (20.7.18) as

groundwater levels fell, though there was a brief response to the July rainfall that resulted in temporary discharge (27.7.18).

### 4.2   Dynamics in stable isotopes

The isotopic samples obtained from different water cycle compartments are displayed in two-dimensional isotope space (Figure 3) supplemented by the global meteoric water line (GMWL), (Craig, 1961). Statistical characteristics are summarised in Table 4. Daily precipitation showed the highest range from being depleted in heavy isotopes ($\delta^{18}$O= -18.3‰; $\delta^2$H= -140.2‰) in

winter to less depleted and even enriched in oxygen relative to the VSMOW standard in summer ($\delta^{18}$O = 0.3‰; $\delta^2$H = -7.3‰). Throughfall samples showed a smaller range in $\delta^{18}$O (min = -17.0‰, max = -1.1‰) and $\delta^2$H (min = -129.5‰, max = -14.9‰). Summer interception derived from 5 throughfall samplers under the forest (11.7.-29.8.2018) averaged to 7 %. In general, precipitation and throughfall samples plotted close to the GMWL. Only exceptionally heavy samples of rainfall and the five associated throughfall samples which originate from one event (4.2 mm) in early August 2018 (8.8.19) showed pronounced

deviations. Spatially distributed precipitation samples in the catchment (not in the plot) showed similar deviations for this event.

The forest and grassland soil samples exhibited substantial damping in isotopic variability relative to precipitation. They also displayed deviations from the GMWL at the more enriched end of their dual isotopic spectrum. We found that water in forest soils had a heavier isotopic composition (Table 4) than grassland soils and more frequent deviations from the GMWL. Monthly

groundwater samples showed the smallest variation with ranges in $\delta^{18}$O (-8.5‰ to -7.4‰) and $\delta^2$H (-56.9‰ to -54.8‰).





Isotopic dynamics in daily stream water were also limited and very similar to groundwater, varying between -8.7‰ & -7.4‰ for $\delta^{18}$O and between -60.6‰ & -51.7‰ for $\delta^2$H and plotted close to the GMWL.

The temporal dimension displayed in in Figure 4 includes daily precipitation amounts, $\delta^{18}$O (a) and $\delta^2$H (b) signatures in precipitation (daily, red) and amount weighted mean throughfall (weekly, green) samples. A precipitation amount-weighted
running mean of the 30 previous days is also shown. The higher sample number and greater temporal resolution of precipitation compared to throughfall (see table 4) resulted in more variability of the isotopic signatures. Open precipitation and throughfall showed seasonal dynamics with more enriched values in both isotopes during the summer and more depleted signatures during the winter. The isotopic variations during summer were dominated by single events with high precipitation amounts. The uneven event distribution led to unsteady running means during the drought. More frequent rain events, between December
and February, lead to increased robustness and congruence of the running means for both isotopes in open precipitation and under canopy.

Heatmaps (Figure 5) show the changing bulk soil water isotope signatures for both landuse types, which are strongly influenced by precipitation inputs (shown in Figure 4). The geometric mean ($\overline{m}_g$) of the replicate samples is displayed as a colour code for all sampled depths. Sampling started in September 2018 after several months of high air temperature and low precipitation
when the severity of the drought became clearer (Figure 2). Under both land use types, the upper 20 cm showed highest variability in isotopic signatures throughout the seasons. This resulted in high SD for both isotopes and sites over all sampling dates at 2.5 cm. The SD decreased with depth (Table 5). During the October sampling, both sites still showed an enriched signal in the top 20 to 30 cm; at both FA and GS, $\delta^{18}$O and $\delta^2$H decreased with depth down to 50 cm. This pattern was not reversed until February, where increased winter precipitation added more depleted water in the upper profile and isotope signatures were
relatively more enriched at depth. Drought impacts on the soil bulk water by evaporation are shown by more negative values of d-excess as this metric was originally presented as an index for non-equilibrium conditions (Dansgaard, 1964). Low values in d-excess are present during the late summer and are most pronounced in the top soil, penetrating the profiles to depths of 30 cm (FA, October) and 20 cm (GS, October) respectively. The enriched October drought signal in both isotopes for the FA and GS top soils persisted and subsequently penetrated downwards, though was damped as evident in the December sampling
at 20-30 cm depth. This pattern was not visible for d-excess values; by December, variations in d-excess over the depth profile were limited under FA and GS.

Estimates of water ages and young water fractions (i.e. % of water younger than 2-3 months) are displayed in Figure 6, and show striking differences with depth between FA and GS soil profiles. The FA young water fraction was > 75 % in the upper 5 soil depths and dropped dramatically to < 5 % at 90 cm depth. GS young water fractions declined more gradually with depth.
Values ranged from > 75 % at the upper most soil layer down to ∼20 % at 90 cm. The method returned significant sine wave fits (Table 6) with the exception of the lowest investigated FA soil layer. The MTT of water at different depths at FA and GS showed similar variations ranging from less than a week in the top soil to 15-20 weeks at 40 to 60 cm. Again, the fits indicated by the KGE were quite good, apart from the very upper and lower soil layers at FA. MTTs in the upper 50 cm were generally longer in GS. The increase in MTT with depth was more exponential in the FA soil producing the most marked difference in
MTT between the sites at 90 cm. While the MTT in the deeper forest soil was above 50 weeks, the GS only showed a MTT of





23 weeks. The shape of the transit time distributions spatially varied from smaller calibrated alpha values in the upper soils to higher alpha values in lower soils at both sites. This reflects more exponential distributions of more rapid transit times fitted in the upper soil and slower, more piston flow-type distribution in the underlying depths.

Several soil layers returned the limits of the manually set parameter space for $\alpha$, $\beta$, or both. Especially the top two layers of GS and three of the FA soil showed low values resulting in MTT being less than a month. This underlines the preliminary nature of this analysis; despite this, the between and within-profile differences were insightful and broadly consistent with the more qualitative insights revealed in Figure 5.

## 5 Discussion

### 5.1 Drought and soil water storage

The climatic anomaly of 2018 provided us with the opportunity to study the impact of drought on different water cycle components of the Demnitzer Millcreek catchment, particularly on soil water storage. Extensive personal communication with local weather-dependent farmers indicated that drought stress in the DMC catchment left insufficient soil moisture to sustain crop demands and green water fluxes in cultivated areas, resulting in losses of agricultural yield that were up to 40 %. These numbers are applicable to the wider region of NE Germany. The extreme conditions of the drought were reflected by meteorological anomalies (SPI, Table 1) and ecohydrological feedbacks (Figure 2). The drought was characterised by low precipitation input (Figure 2 a), high temperatures (a), low soil water storage (b, c), declining groundwater levels (d), and stream flow ceasing (d). Only the heavy rainfall in July prevented the drought being more severe, given the persistence of dry warm weather into the late autumn. We found that local observations in DMC were consistent with other, large scale observations on the characteristics of the drought (Imbery et al., 2018). We observed differences of the drought dynamics under the FA and GS plots which were expected considering the importance of vegetation on water cycling in the critical zone (e.g., Dubbert and Werner, 2019). Generally, soils under the forest were drier than the grassland, which likely reflects the greater interception and transpiration losses under forest that have been observed elsewhere in Brandenburg (Douinot et al., 2019). However, it is clear that differences in soil properties also result in greater moisture storage and retention in the more clay-rich upper profile of the grassland soil. Dynamics of PET and transpiration rates of oak trees (Figure 2) imply that the trees were able to successfully sustain transpiration throughout this drought, as transpiration rates did not appear to be reduced relative to atmospheric demand (indexed by PET). Low soil water availability under the forest during drought conditions raise the question of the origin of the water transpired by oaks. Clearly, the high rainfall inputs in July, which replenished storage in the top soil were likely critical in enabling transpiration to continue through July and August via rapid recycling. However, oak trees may also be able to access deeper water sources (e.g., David et al., 2004) near the water table via deep roots (which were present at 1 m in the FA sampling plot). In addition, ecophysiologically based water-limitation-tolerance has been observed in various oak species by Hahm et al. (2018).





## 5.2 Insights from stable isotopes

Isotopes were key tools used in this study to assess the impact of the drought on different soil- vegetation assemblages. The direct equilibrium method applied used monthly destructive soil sampling to return stable isotope ratios in the soil bulk water molecules. This provided further insight in spatio-temporal patterns of water movement and mixing in the unsaturated zone (Figure 5). Using heat maps, we were able to visualise qualitative patterns of site-specific advection and dispersion of the isotopic input signal from the soil-atmosphere interface down to 1m. Further, the evaporation signal from the drought 2018 was apparent in the soil bulk water d-excess profiles at the forested and grassland site in September and October. Summer soil evaporation, in combination with precipitation deficits during the drought, were likely the driving processes leading to the temporarily enriched (compared to recent precipitation) isotope signatures in the topsoils for September and October at both sites. Both sites showed a subsequent displacement and mixing of the bulk soil isotope signatures with incoming precipitation as re-wetting progressed in the autumn and early winter. However, the evolution of isotopic and d-excess signal in the unsaturated soil storage indicates differences between sites (Figure 5). The forest soil being sandier and having little direct ground vegetation cover, allowed a deeper penetration of the evaporation front and therefore more negative d-excess values in depths up to 30 cm. Limited transfer of d-excess effects to depths observed in this study is in accordance with findings of Sprenger et al. (2016). Lower soil moisture (and therefore water storage) in the forest (Figure 2) likely led to stronger Rayleigh fractionation effect on the remaining bulk soil water. Mixing with incoming precipitation resulted in moisture replenishment and rapid over-printing of the isotopic drought signal (d-excess) at both sites. We did not see a strong "memory"-effect of the d-excess in individual soil depths on a monthly time step over the entire study period. The apparent contradiction between our findings and recent findings by (Sprenger et al., 2019a) who reported consistently disjunct soil water pools, likely simply reflects different soil and climatic properties, leading to different hydrological pathways. (Sprenger et al., 2018) described the bound soil water as water under different matric potentials. This has implications for the interpretation of interactions speeds between different water pools in the soil and their partitioning into green and blue water fluxes. Findings by Bowers et al. (2020) support the relatively fast time-dependent isotopic mixing of water held under different tensions in the matrix of sandy soils. A study in a controlled ecosystem (Evaristo et al., 2019) further highlights the importance of spatio-temporal dynamics in soil water for its partitioning and interpretation of resulting patterns.

## 5.3 Soil water response and travel times

We were also able to use the isotope time series to provide a first approximation of the travel times of water in the soil during the re-wetting phase. Despite the short data time-series, fitting simple sine wave and gamma models (Figure 6) provided useful insights into the spatial differences in the young water fraction and MTTs between, and within, the soil profiles of the two sites. As we sampled bulk soil water at only monthly intervals, the resulting values can only be regarded as indicative, but they capture the shift from dry to wet conditions. The upper soils are dominated by younger water (less than a month) at both sites and age increases with depth (Figure 6). Likely causes of the poor model fits of TTD in the upper soil are evaporative fractionation and the sampling frequency being too coarse to capture the temporal resolution of these processes in the sandy



soils. Furthermore, the bulk soil water sampling likely underestimates the effects of preferential flow, especially in the more
320  heterogeneous upper horizons. Nevertheless, similar ages and differences were reported by Smith et al. (submitted) from a
physically based tracer-aided ecohydrological modelling approach; this increases confidence that the results are instructive.
We found a steady increase in age with depth at the grassland site and generally higher soil moisture content. These patterns
were represented by higher $\alpha$-values in comparison to the FA site (Table 6) which implies a more piston-like flow through the
soil profile. Similar patterns of increasing alpha values due to enhanced dampening with depth in the profile were reported by
325  (Tetzlaff et al., 2014) in freely draining podzols in the Scottish Highlands. This leads to more consistent moisture flux to depth,
which is characteristic of the grassland site. In contrast, the FA soil was more freely draining, and drier, and hence younger
water could have a greater influence following rainfall events, even in the deeper soil layers. Substantially older soil water at
80 to 100 cm depth indicates more irregular groundwater recharge at the FA site, though the poor model fit here highlights
the low variability in the isotopic composition of the deep forest soil and resulting uncertainty. These tracer-based inferences
were supported by the soil moisture dynamics (Figure 2 b & c) which also indicated more frequent percolation to depth under
grassland and a temporally limited (to winter) groundwater recharge under forest. Deeper soil bulk water that mainly represents
groundwater recharge was older under the forested plot. This is in accordance with findings from tracer-aided modelling of
water age dynamics under forest and grassland at another site in Brandenburg from Douinot et al. (2019). The displacement
of the isotopic signal down the soil profile was not observed in d-excess, with limited "memory effect" of the drought 2018
with time and depth in both soils. But, despite the rapid recovery of the d-excess signal in the isotopic composition of bulk
soil water, this was not indicative of drought recovery. With SPI values still low for longer averaging periods, the effects of the
drought impact were still evident in the catchment. Stream flow did not recover until the beginning of 2019, and even then,
flows were subsequently much lower than the previous winter (Figure 2). The dual-isotopic characteristics of groundwater
(Figure 3) suggest a well-mixed storage. The concurrence of the isotopic composition of this storage with the stream signal
indicates that the DMC is a groundwater fed stream. This is also consistent with recent analysis of the catchment flow data
(Smith et al., 2020), and is further supported by the temporal synchrony of stream flow reoccurrence and groundwater recovery
(Figure 2 d). The drought intensity value (Table 2) still indicates a substantial deficit in soil and groundwater stores reflecting
incomplete rewetting of the catchment at the end of the study period.

## 5.4  Wider implications

Sandy soil properties and weather-dependent farming make landscapes like the DMC vulnerable to droughts in a continental
hydroclimate where dry, hot summers are likely to become more common. Understanding and managing soil moisture in the
unsaturated zone of the catchment will be fundamentally important to developing land use strategies that will be resilient in
the face of climatic warming. Both crops and trees (Amin et al., 2019) primarily rely on shallow soil moisture storage and
there is a clear trade-off between biological productivity, water use and the maintenance of other ecosystem services such
as groundwater recharge and river flow generation. It is important that management of crop lands and forests is based on an
understanding of how water is partitioned into green and blue water fluxes, so that evidence-based decisions can be made that
prioritise water use for the most important, sustainable societal needs. The ephemeral character of the DMC stream – which has





been perennial in the past – during the drought, underlined the failure in supporting blue water fluxes that can be of importance
for habitat structure, connectivity and water quality. For example, recent recolonization of the catchments by beavers (Castor

fiber) has had a major impact on flow regimes and water quality; changes that might not be sustained if the stream becomes
ephemeral for longer periods (Smith et al., 2020). The need for greater understanding of water security is given urgency with
the by alarming climate projections for the area (Lüttger et al., 2011). Climate change impacts are already cross-sectorally
perceived by local land use managers in the North German Plain (Barkmann et al., 2017). The catchment failed to maintain
normally expected green and blue water fluxes throughout the growing season and drought of 2018 which made ecosystem

services and agricultural land use unsustainable. To improve system understanding and management strategies, further research
is needed. The range of different land use types / soils has to be broadened to capture large scale heterogeneity. Further, the
obtained datasets have to be integrated in models to enable quantitative estimations, upscaling to larger areas and extrapolation
in time and climatic contexts of these processes. On the basis of our work here, we would argue that insights from stable
isotopes could play a key role in this process. To exploit the potentials of stable water isotopes to gain process insight, studies

on finer spatial-temporal resolutions are fundamental. Those field based assessment in a natural setting are the basis to further
evolution of stable isotopes as tracers in hydrology. We propose that these studies should be hand in hand with methodological
and model development.

## 6 Conclusions

We presented an assessment of the 2018 drought and associated ecohydrological feedbacks in a lowland headwater in North

East Germany (DMC) using hydroclimatic data in conjunction with isotope-based techniques. The study focused on two plot
sites with differing vegetation/soil communities during a period of water stress, when the catchment could no longer sustain
blue water fluxes (e.g. stream flow) or green water needs (e.g. crop production) and the subsequent recovery. In general, the
forest site "used" more water and was more freely draining and hence had drier soils. Nevertheless, the transpiration dynamics
of Oak trees showed some resilience towards drought conditions and appeared to meet atmospheric moisture demand. We

derived insight into the subsurface unsaturated storage dynamics by monitoring isotope ratios in bulk soil. This destructive
sampling showed spatio-temporal distinctions in isotope distributions and water ages between sites. Water in the upper soil
profile was dominated by relatively recent rainfall (< 2-3 months age), with age increasing with depth to > 6 months. The
deeper forest soil horizons appear to have only old water (∼ year old). In contrast to the individual isotopic signature of soil
water, which took some time to be erased by mixing with winter rainfall, no "memory"-effect or displacement of the drought

signal with depth was observed in d-excess. Re-occurring of precipitation at the end of the study period revealed differences
in ground water recharge dynamics between plot sites, which was lower and more intermittent under the forest site. These
patterns have potential implications for blue and green water management in such environments and should be investigated in
a greater range of representative vegetation/ soil units. This highlights the need for further research efforts on climate change
and management adaptations in the critical zone of drought sensitive lowland ecosystems.



*Data availability.* The data from the study are available from the corresponding author upon request.

*Author contributions.* DT, CS and LK designed the study. HW set up the sap-flow measurements and AWS and helped with analysis. LK collected the data and carried out the fieldwork. AS helped with MTT analysis. LK performed the analysis and prepared the first draft of the manuscript. All the authors edited and commented on the manuscript.

*Competing interests.* The authors declare that they have no conflict of interest.

*Acknowledgements.* This work was partly funded by the European Research Council (project GA 335910 VeWa). We acknowledge support by the German Research Foundation (DFG) and the Open Access Publication Fund of Humboldt-Universität zu Berlin. The authors are grateful to all colleagues involved in the sample collection (in particular A. Wieland, N. Weiß, W. Lehmann), D. Dubbert for support with the isotope analysis, as well as T. Rossoll for help with the measurement equipment. L. Lachmann helped with the GIS data and maps. We are thankful for on-site support and knowledge by the WLV (Wasser- und Landschaftspflegeverband Untere Spree). M. Facklam (TU Berlin), N.
Rosskopf and A. Markert (both LBGR Brandenburg) are also thanked for their help in soil sampling.



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

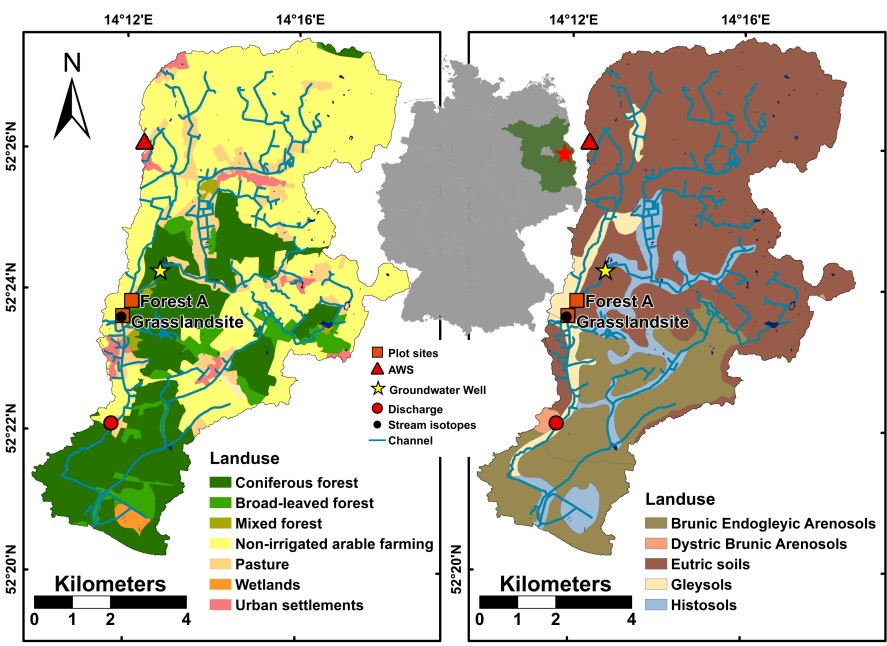

**Figure 1.** Study site location, sampling site locations, soils and landuse (Source: ©GeoBasis-DE/BKG 2018; data changed) of the DMC catchment.



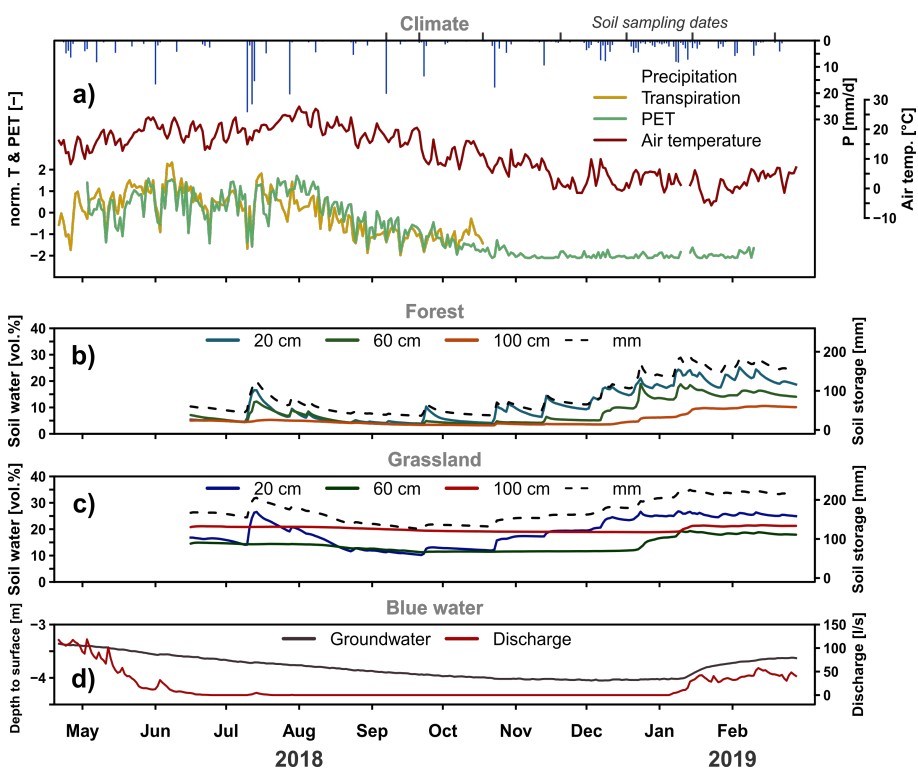

**Figure 2.** Daily precipitation, normalized transpiration (from sap-flow sensors), normalised PET (FAO Penman-Monteith) (a), volumetric soil moisture (n = 6 per 3 depths of the forested plot (b) and grassland site (c) with the total soil water in mm (- -) and ground water elevation relative to surface and discharge at the Demnitz mill (d) for the growing season 2018 and rewetting 2019.

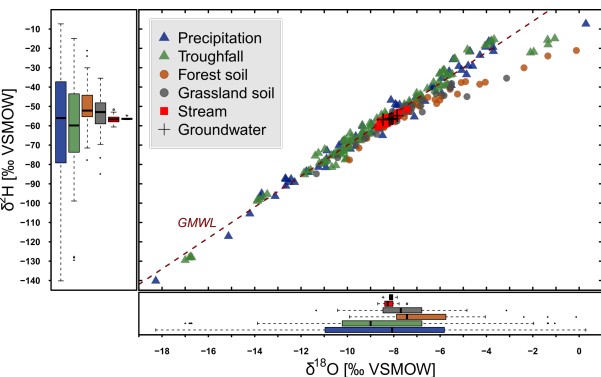

**Figure 3.** Dual isotope plot of DMC precipitation, throughfall, forest and grassland soil bulk water samples, as well as stream and groundwater.





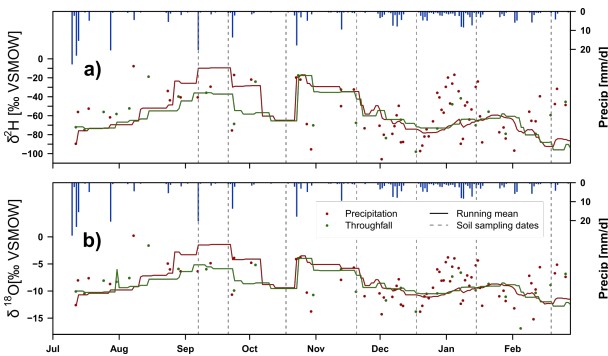

**Figure 4.** Time series of daily precipitation amount and isotopic signature of weekly and daily precipitation and their 30 days running (volume weighted) mean for $\delta^2$H (a) and $\delta^{18}$O (b) with dotted lines indicating dates of bulk soil water sampling (- -).

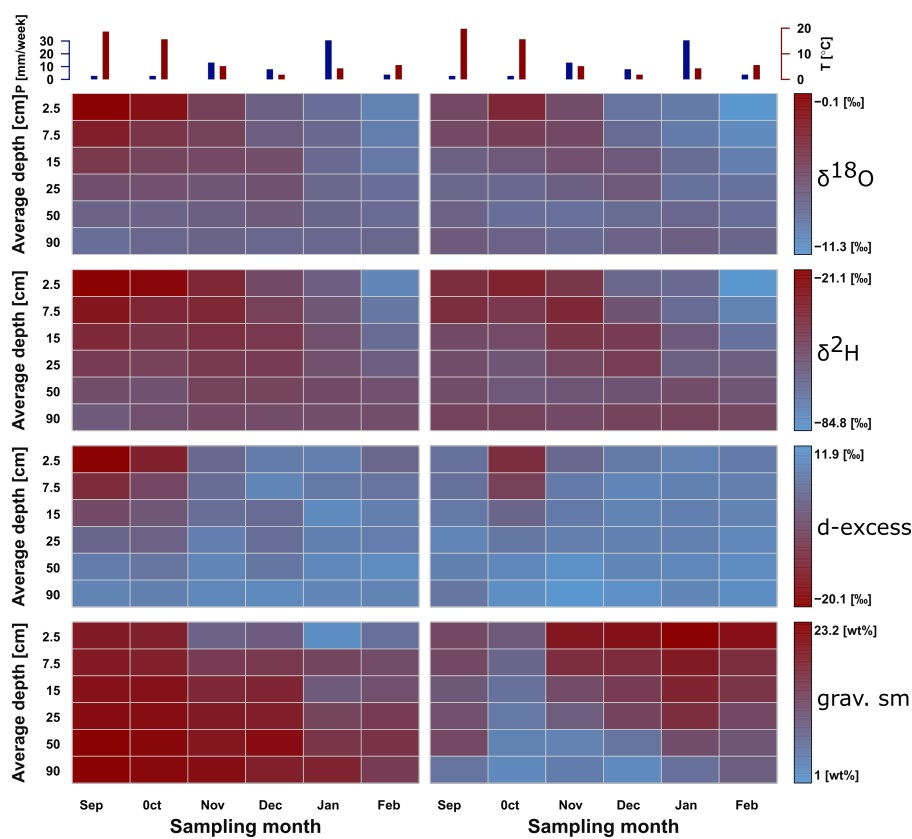

**Figure 5.** Heat map of soil depth profiles for mean $\delta^2$H, $\delta^{18}$O, d-excess and grav. soil moisture of three replicates at both land use classes (forest & grassland) of six monthly sampling dates. Blue to red colours indicate more to less depleted $\delta^2$H & $\delta^{18}$O values, declining d-excess, and grav. soil moisture values. On top, antecedent conditions of mean daily air temperature and sum of precipitation 7 days prior to sampling are displayed in the bar chart.



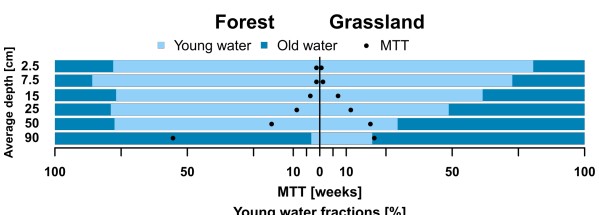

**Figure 6.** Young water fractions and MTT per soil depths at forested and grassland site.



**Table 1.** Soil characteristics of the grassland and forest plot sites (sampled on 20.3.2019).

| Depth | Clay | Silt | Sand | dry bulk density | pH | carbonate | ignition loss | N | C | TOC | C/N |
|---|---|---|---|---|---|---|---|---|---|---|---|
| top - bottom [cm] | < 0.002 [mm] | 0.002-0.063 [mm] | 0.063-2.0 [mm] | [g/cm$^3$] | [-] | [wt %] | [wt %] | [wt %] | [wt %] | [wt %] | [-] |
| **Grassland** | | | | | | | | | | | |
| 0 - 8 | 6.3 | 11.3 | 82.4 | - | 4.3 | 0.0 | 4.6 | 0.3 | 2.4 | 2.5 | 8.0 |
| 8 - 28 | 7.7 | 11.0 | 81.3 | 1.3 | 5.5 | 0.0 | 4.2 | 0.3 | 2.6 | 2.5 | 8.7 |
| 28 - 42 | 3.8 | 8.6 | 87.6 | 1.4 | 5.8 | 0.0 | 2.6 | 0.1 | 1.3 | 1.3 | 13.0 |
| 42 - 70 | 1.0 | 1.6 | 97.3 | 1.5 | 6.0 | 0.0 | 0.2 | < 0.1 | < 0.5 | < 0.5 | - |
| 70 - 95 | 0.8 | 0.4 | 98.8 | - | 5.9 | 0.0 | 0.2 | < 0.1 | < 0.5 | < 0.5 | - |
| **Forest** | | | | | | | | | | | |
| 0 - 5 | 3.2 | 13.0 | 83.7 | - | 3.4 | 0.0 | 5.2 | 0.1 | 3.0 | 2.9 | 30.0 |
| 5 - 8 | 3.7 | 12.2 | 84.1 | 1.0 | 3.4 | 0.0 | 7.1 | 0.2 | 4.0 | 3.9 | 20.0 |
| 18 - 35 | 1.3 | 9.6 | 89.1 | 1.4 | 3.6 | 0.0 | 1.8 | < 0.1 | 0.9 | 0.9 | - |
| 35 - 65 | 1.9 | 5.0 | 93.1 | 1.0 | 4.0 | 0.0 | 0.7 | < 0.1 | < 0.5 | < 0.5 | - |
| 65 - 70 | 8.9 | 8.0 | 83.2 | - | 5.3 | 0.0 | 1.3 | < 0.1 | < 0.5 | < 0.5 | - |
| 70 - 120 | 7.3 | 3.1 | 89.6 | - | 8.1 | 59.3 | 3.4 | < 0.1 | 7.4 | 0.7 | - |





**Table 2.** SPI meteorological drought intensity values for different periods derived from historical precipitation distribution.

| SPI | Jan | Feb | Mar | Apr | May | Jun | Jul | Aug | Sep | Oct | Nov | Dec | Jan | Feb |
|---|---|---|---|---|---|---|---|---|---|---|---|---|---|---|
| **1 month** | 1.1 | -2.1 | 0.6 | -0.3 | -1.4 | -1.0 | 0.9 | -1.3 | -1.0 | -0.5 | -1.2 | 0.0 | 0.2 | -0.4 |
| **3 month** | 1.0 | -0.2 | 0.4 | -0.7 | -0.9 | -1.9 | -0.6 | -0.6 | -0.4 | -1.8 | -1.9 | -1.2 | -0.8 | -0.3 |
| **6 month** | 0.7 | 0.4 | 0.8 | 0.2 | -0.8 | -1.2 | -1.0 | -1.2 | -1.6 | -1.6 | -1.6 | -1.0 | -1.7 | -1.5 |
| **9 month** | 1.3 | 1.2 | 0.8 | 0.3 | -0.2 | -0.4 | -0.4 | -1.1 | -1.3 | -1.8 | -1.9 | -2.0 | -1.8 | -1.5 |
| **12 month** | 1.2 | 1.0 | 1.0 | 1.1 | 0.7 | 0.0 | -0.2 | -0.6 | -0.7 | -1.2 | -1.8 | -1.6 | -2.0 | -1.8 |



**Table 3.** Summary statistics for the daily time series shown in Figure 2.

| Parameter | Mean | Median | Max | min | SD | Unit |
|---|---|---|---|---|---|---|
| **Climate** | | | | | | |
| Precipitation | 1.2 | 0.0 | 27.0 | 0.0 | 3.4 | mm/d |
| Temperature | 11.9 | 12.9 | 27.7 | -5.7 | 8.2 | °C |
| Transpiration | 0.0 | 0.1 | 2.3 | -2.0 | 1.0 | mm/d |
| PET | -0.8 | -1.2 | 1.7 | -2.1 | 1.2 | mm/d |
| **Forest** | | | | | | |
| VWC 20 cm | 11.1 | 9.1 | 25.2 | 3.9 | 6.6 | $m^3/m^3$ |
| VWC 60 cm | 8.2 | 5.8 | 18.9 | 3.5 | 4.7 | $m^3/m^3$ |
| VWC 100 cm | 5.3 | 4.5 | 10.6 | 3.2 | 2.3 | $m^3/m^3$ |
| Storage | 87.7 | 66.4 | 187.4 | 36.3 | 48.4 | mm |
| **Grassland** | | | | | | |
| VWC 20 cm | 18.9 | 18.9 | 26.7 | 10.2 | 5.4 | $m^3/m^3$ |
| VWC 60 cm | 14.0 | 13.4 | 19.3 | 11.3 | 2.6 | $m^3/m^3$ |
| VWC 100 cm | 20.0 | 20.1 | 21.5 | 18.9 | 0.9 | $m^3/m^3$ |
| Storage | 171.7 | 166.1 | 225.6 | 125.0 | 30.6 | mm |
| **Blue water** | | | | | | |
| Discharge | 15.8 | 0.0 | 118.7 | 0.0 | 28.2 | l/s |
| Groundwater | -3.8 | -3.8 | -3.4 | -4.1 | 0.2 | m |



**Table 4.** The number (n), 50[th] (median), 5[th] and 95[th] percentiles of $\delta^{18}$O & $\delta^2$H [‰VSMOW] signatures of the sampled water cycle compartments.

| | | $^{18}$O [‰ VSMOW] | | | $^2$H [‰ VSMOW] | | |
| --- | --- | --- | --- | --- | --- | --- | --- |
| | | | percentile | | | percentile | |
| | n | Median | 5$^{th}$ | 95$^{th}$ | Median | 5$^{th}$ | 95$^{th}$ |
| **Precipitation** | 68 | -8.1 | -13.7 | -3.8 | -56.0 | -96.7 | -19.3 |
| **Throughfall** | 136 | -9.0 | -13.7 | -3.8 | -59.9 | -97.0 | -18.5 |
| **Forest soil** | 36 | -7.4 | -9.2 | -2.1 | -52.2 | -67.7 | -28.5 |
| **Grassland soil** | 36 | -7.7 | -9.8 | -5.3 | -52.9 | -71.3 | -39.9 |
| **Stream** | 51 | -8.3 | -8.6 | -8.6 | -56.7 | -59.1 | -53.7 |
| **Groundwater** | 6 | -8.1 | -8.4 | -7.9 | -56.4 | -56.8 | -55.2 |





**Table 5.** Descriptive statistics (mean, median, standard deviation (SD) and percentiles (5[th], 95[th])) for the geometric mean (n=3) of the soil isotopes samples $\delta^{18}$O & $\delta^2$H [‰ VSMOW] values per sites and depths for all sampling dates (n=5).

| Depth [cm] | | $\delta^{18}$O [‰ VSMOW] | | | Percentile | | $\delta^2$H [‰ VSMOW] | | | Percentile | |
|---|---|---|---|---|---|---|---|---|---|---|---|
| | | Mean | Median | SD | 5[th] | 95[th] | Mean | Median | SD | 5[th] | 95[th] |
| **Forest** | 2.5 | -5.3 | -6.1 | 3.6 | -9.5 | -0.4 | -45.5 | -44.8 | 20.2 | -74.2 | -26.6 |
| | 7.5 | -6.0 | -6.1 | 2.4 | -9.1 | -2.8 | -47.2 | -43.1 | 14.1 | -68.8 | -37.5 |
| | 15 | -6.4 | -5.8 | 1.7 | -8.9 | -4.5 | -48.3 | -44.2 | 9.7 | -64.2 | -41.7 |
| | 25 | -6.8 | -6.4 | 0.9 | -8.3 | -6.0 | -50.5 | -47.6 | 5.6 | -59.7 | -45.5 |
| | 50 | -7.5 | -7.5 | 0.4 | -8.0 | -7.0 | -52.7 | -53.1 | 2.5 | -55.6 | -49.3 |
| | 90 | -7.8 | -7.8 | 0.2 | -8.2 | -7.6 | -54.6 | -54.1 | 2.3 | -54.9 | -52.2 |
| **Grassland** | 2.5 | -7.3 | -7.3 | 2.7 | -10.8 | -3.7 | -55.9 | -54.4 | 17.3 | -80.1 | -36.7 |
| | 7.5 | -7.3 | -6.8 | 2.1 | -10.1 | -5.0 | -53.9 | -50.7 | 14.1 | -74.1 | -38.7 |
| | 15 | -7.5 | -7.0 | 1.2 | -9.2 | -6.3 | -54.1 | -52.7 | 8.6 | -67.0 | -44.3 |
| | 25 | -7.8 | -7.8 | 0.7 | -8.6 | -6.9 | -55.2 | -55.7 | 5.2 | -61.4 | -48.0 |
| | 50 | -8.1 | -8.2 | 0.4 | -8.5 | -7.6 | -56.1 | -56.8 | 2.1 | -58.4 | -53.1 |
| | 90 | -7.6 | -7.5 | 0.4 | -8.0 | -6.9 | -50.3 | -49.7 | 1.2 | -52.1 | -49.3 |



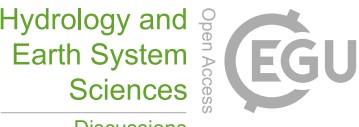

**Table 6.** Soil bulk water MTT estimates, including their $\alpha$ - and $\beta$ -value, from best fits of gamma function and young water fraction from best sine wave fit and associated p-value.

|  | Depth [cm] | MTT MTT [month] | $\alpha$ | $\beta$ | KGE | Young water fraction ywf [%] | p-value |
|---|---|---|---|---|---|---|---|
| **Forest** | **2.5** | < 1 | 0.5 | 2.0 | 0.15 | 78 | 4E-02 |
|  | **7.5** | < 1 | 0.5 | 2.0 | 0.50 | 86 | 1E-03 |
|  | **15.0** | < 1 | 1.8 | 2.0 | 0.66 | 77 | 3E-04 |
|  | **25.0** | 2.1 | 2.6 | 3.5 | 0.80 | 79 | 1E-04 |
|  | **50.0** | 4.4 | 1.9 | 9.9 | 0.87 | 31 | 9E-06 |
|  | **90.0** | 13.7 | 1.1 | 52.0 | 0.01 | 3 | 5E-01 |
|  |  |  |  |  |  |  |  |
| **Grassland** | **2.5** | < 1 | 0.5 | 2.0 | 0.69 | 81 | 3E-02 |
|  | **7.5** | < 1 | 0.8 | 2.0 | 0.79 | 73 | 3E-03 |
|  | **15.0** | 1.8 | 2.7 | 2.9 | 0.89 | 62 | 2E-05 |
|  | **25.0** | 3.0 | 2.5 | 5.2 | 0.87 | 49 | 4E-04 |
|  | **50.0** | 4.8 | 2.4 | 8.6 | 0.74 | 29 | 7E-04 |
|  | **90.0** | 5.2 | 3.4 | 6.6 | 0.51 | 20 | 5E-02 |