# Peer review of "Using water stable isotopes to understand evaporation, moisture stress and re-wetting in catchment forest and grassland soils of the summer drought of 2018."

_Hydrology and Earth System Sciences, 2020_

## Referee Comment (RC1) · Anonymous Referee #1 · 3 Apr 2020

General comments

In the manuscript entitled 'Using isotopes to understand evaporation, moisture stress and re-wetting in catchment forest and grassland soils of the summer drought of 2018' the question is addressed, how the drought 2018 affected two different land-use/soil community sites in a catchment in NE Germany. The two plot sites were monitored during a period of water stress, when the catchment could no longer sustain blue water fluxes (e.g. stream flow) or green water needs (e.g. crop production), and the subsequent recovery.

Ecosystem response to this climatic anomaly is investigated by using water stable isotope data of precipitation and throughfall, stream-, groundwater and especially from soil water profiles. Monthly soil profile samples in 6 different depths down to 1 m under two different land-use types were taken from September 2018 to February 2019. Soil water isotopes were analysed using direct vapor equilibration laser spectrometry (DVE-LS). These data were used to estimate mean transit times (MTT) in the soils at the different depths as well as young water fractions, using a fitted sine-wave method. Based on collected meteorological and sap flow data ET-pot was calculated. Soil moisture was monitored in three different depths at both sites. Drought severity was quantified with the SPI, based on long-term precipitation data from the DWD.

It could be shown that the forest soils were dominated by rapid young water fluxes after rainfall events whereas the grassland soils were more retentive and dominated by older water. It is concluded that implications for blue and green water management should be investigated in a greater range of representative vegetation/ soil units and that further research efforts on climate change and management adaptations in the critical zone of drought sensitive ecosystems is needed.

Overall, the manuscript is well structured and nicely written.

The topic fits well to the scope of the journal and appears to be of interest for the readers; I only suggest moderate revisions prior to acceptance and publication in Hydrology and Earth System Sciences.

Specific comments

Title

Please add which isotopes were investigated, either 'water stable isotopes' or 'stable isotopes (d18-O, d2-H)'.

Evaporation or Evapotranspiration?

L. 4

Please add 'water' before "...stable isotopes to..."

L. 47

Compared to e.g. soil moisture probes, laser absorption spectroscopy is not really "inexpensive"... I wouldn't go too deep into the history of stable isotope measurement techniques, but it could be added that compared to traditional mass spec. techniques, laser absorption spectroscopy is relatively inexpensive. Mentioning that the invention of laser absorption spectroscopy has facilitated several new techniques in the last ten years would also emphasise that your approach is relatively new.

L. 52

Developed by whom? Not by Hendry et al., maybe they improved. Please delete "e.g.," in the brackets.

L. 60

Shift "To" to line 59 after "study"

L. 66

'located' instead of "based"

L. 123

Throughfall was sampled as well at 1 m height? Which distance between the five gauges?

L. 131

blank is missing between "for" and "d2H"

L. 132

Please complete "...from October 2018 to..."

L. 143

Please provide part-no. of sample bags

L. 146

Please insert 'gas' after "headspace"

L. 147

What kind of 'standards' were used?

L. 150

d-Excess should be introduced somewhere here.

L. 150-152

Method or results section?

L. 157

Please add 'oil' after "paraffin"

L. 171

Where can I see "young water" in figure 2?

L. 171-172

Method or results section?

L. 213

One would always expect slightly evaporated signals in throughfall (enriched in heavy isotopes) compared to precipitation. In your study it is opposite (Fig. 4, Table 4), this seems to be in contradiction to your soil profile data (Table 5). Please elaborate on this.

L. 223

Please delete one of the "in" before "Figure 4"

L. 225

Is the sample number of precipitation really higher than throughfall? According to Table 4: precipitation (68), throughfall (136). Please clarify.

L. 232

I like the heat map (Figure 5), but you could think about providing a figure for each site with the soil profile isotope data as supplemental material.

L. 238-240

Not clear what you mean, please rephrase.

L. 259

'upper' instead of "top"

L. 260

Please insert 'the upper' before "three of the..."

L. 289

Insight 'into' instead of "in"

L. 316

Please replace "soils" by 'soil layers'.

L. 324

Stick to "ðİđł-values" to be consistent with L. 323 and Table 6.

L. 338

Please insert 'in' or 'reported from' before "...the previous winter".

L. 342

'storage,' instead of "stores"

L. 356-357

"...urgency with the by..." ??? Please rephrase.

L. 369

"...headwater..." is 'catchment' missing?

L. 376

Please add 'the two' between "between" and "sites"

L. 378

Please insert '1' before "year old)"

Fig. 1

There is enough space to put the overview beneath the detailed maps – makes the layout of the figure a bit clearer.

Right part: please replace "Landuse" by 'Soils'

Fig. 3

Legend: 'h' is missing in "Troughfall"

Fig. 5

Please label forest and grassland.

Table 4

Deltas are missing in header

Are 5th and 95th percentile of d18-O identical for precipitation and throughfall? Stream:

5th and 95th percentile of d18-O both -8.6? Please double-check.

Table 5

Caption: 'soil water isotope samples' instead of "soil isotopes samples"

Global changes

I would prefer "and" instead of "&" (e.g. l. 62, 112, 136)

please change "stable water isotopes" to 'water stable isotopes'

check '-' vs. '–' throughout the manuscript (e.g. L. 287, L. 291)

Please also note the supplement to this comment:
https://www.hydrol-earth-syst-sci-discuss.net/hess-2020-81/hess-2020-81-RC1-supplement.pdf

---

## Referee Comment (RC2) · Anonymous Referee #2 · 28 Apr 2020

The manuscript entitled "Using isotopes to understand evaporation, moisture stress and re-wetting in catchment forest and grassland soils of the summer drought of 2018." by Lukas Kleine, Dörthe Tetzlaff, Aaron Smith, Hailong Wang, and Chris Soulsby presents an interesting contribution to our understanding of ecohydrological processes in a mixed land cover catchment (forest and agricultural), especially under the influence of climate anomalies. The authors conducted a case study in North-East Germany in the Demnitzer Millcreek catchment. They highlight the use of isotopic tracers together with conventional hydrology to understand the effect of drought progress, the recovery

of soil water storage and the memory effect of drought evaporation when the catchment could no longer hold streamflow and crop production and further mixing with fresh precipitation.

The study shows an important work with a logical structure and is clearly written, in my opinion, it deserves to be considered for publication in the HESS after some minimal revisions. Most of my editing comments match those of Referee 1 and have already been addressed by the authors.

I recommend the authors to be careful when using the terms "blue and green water", as it is broad and varied in the literature, so I suggest that they highlight in the introduction section what they specifically refer in this study.

I'm a little concerned about the limited availability of soil water isotope samples (monthly basis) used to drive such a conclusion based on tentative MTTs. The manuscript would benefit for a wider discussion and to clearly state this limitation. In order to reaffirm the credibility of these results, I suggest widening the context of the study by comparing it with similar drought cases in nearby sites or with comparable geographical regions. Further, an extended amount of literature pointed out that MTT (based is a gamma distribution with two parameters and derived MTTs concept) is only a qualitative indicator of catchments for a first screen and basic comparison, however a bit critical when the evolution of water ages is involved. With the available information, I firmly believe that it would be possible to obtain better and accurate results by including more elaborate and non-stationary criteria in the analysis.

Finally, please improve figure 4, the size of the symbols and the colours used make it difficult to identify isotopic signatures.

---

## Author Comment (AC1) · 22 May 2020

*Throughout the revision of the manuscript, the authors have corrected the comments by the reviewers and have addressed them accordingly.*

**Anonymous Referee #1**

General comments

In the manuscript entitled 'Using isotopes to understand evaporation, moisture stress and re-wetting in catchment forest and grassland soils of the summer drought of 2018' the question is addressed, how the drought 2018 affected two different land-use/soil community sites in a catchment in NE Germany. The two plot sites were monitored during a period of water stress, when the catchment could no longer sustain blue water fluxes (e.g. stream flow) or green water needs (e.g. crop production), and the subsequent recovery.

Ecosystem response to this climatic anomaly is investigated by using water stable iso-tope data of precipitation and throughfall, stream-, groundwater and especially from soil water profiles. Monthly soil profile samples in 6 different depths down to 1 m under two different land-use types were taken from September 2018 to February 2019. Soil water isotopes were analysed using direct vapor equilibration laser spectrometry (DVE-LS). These data were used to estimate mean transit times (MTT) in the soils at the different depths as well as young water fractions, using a fitted sine-wave method. Based on collected meteorological and sap flow data ET-pot was calculated. Soil moisture was monitored in three different depths at both sites. Drought severity was quantified with the SPI, based on long-term precipitation data from the DWD.

It could be shown that the forest soils were dominated by rapid young water fluxes after rainfall events whereas the grassland soils were more retentive and dominated by older water. It is concluded that implications for blue and green water management should be investigated in a greater range of representative vegetation/ soil units and that further research efforts on climate change and management adaptations in the critical zone of drought sensitive ecosystems is needed.

Overall, the manuscript is well structured and nicely written.

The topic fits well to the scope of the journal and appears to be of interest for the read-ers; I only suggest moderate revisions prior to acceptance and publication in Hydrology and Earth System Sciences.

*We thank the reviewer 1 for the encouraging comments on our manuscript. We are grateful for this very detailed and careful review of our work.*

Specific comments

Title

Please add which isotopes were investigated, either 'water stable isotopes' or 'stable isotopes (d18-O, d2-H)'.

*We specified this as 'water stable isotopes' in the manuscripts title.*

Evaporation or Evapotranspiration?

*We choose evaporation as the focus of the isotope techniques is on evaporation.*

L. 4
Please add 'water' before ". . .stable isotopes to. . ."

*'water' was added to particularize the isotopes used.*

L. 47

Compared to e.g. soil moisture probes, laser absorption spectroscopy is not really "inexpensive". . . I wouldn't go too deep into the history of stable isotope measurement techniques, but it could be added

that compared to traditional mass spec. techniques, laser absorption spectroscopy is relatively inexpensive. Mentioning that the invention of laser absorption spectroscopy has facilitated several new techniques in the last ten years would also emphasise that your approach is relatively new.

*Thanks for the suggestion. We have added more detail on that.*

L. 52

Developed by whom? Not by Hendry et al., maybe they improved. Please delete "e.g.," in the brackets.

*Thank you. This was indeed very misleading.*

L. 60
Shift "To" to line 59 after "study"

*Thank you. We adjusted it.*
L. 66
'located' instead of "based"

*Changed according to suggestion.*

L. 123

Throughfall was sampled as well at 1 m height? Which distance between the five gauges?

*We added more detail regarding the experimental setup of the throughfall to the revised manuscript.*
L. 131
blank is missing between "for" and "d2H"

*Adjusted. Thank you.*
L. 132
Please complete ". . .from October 2018 to. . ."

*We completed this (to February 2019)*
L. 143
Please provide part-no. of sample bags

*Thank you. Indeed part-no changed in comparison to older studies.*
L. 146
Please insert 'gas' after "headspace"

*Done as suggested.*
L. 147
What kind of 'standards' were used?

*We added more detail to the standards (liquid/ 10ml) but too much detail might be confusing as these standards are only used for calibration and not relevant for the further storyline.*

L. 150
d-Excess should be introduced somewhere here.

*We included a short introduction to the d-excess concept.*

L. 150-152

Method or results section?

*We presented this information here to give insight into the method and its precision.*

L. 157

Please add 'oil' after "paraffin"

*Thank you. We specified it to "paraffin oil"*

L. 171

Where can I see "young water" in figure 2?

*This was a mistake – we removed the reference to Fig 2 here.*

L. 171-172

Method or results section?

*We mentioned this here to clarify the definition of young water in the applied method.*

L. 213

One would always expect slightly evaporated signals in throughfall (enriched in heavy isotopes) compared to precipitation. In your study it is opposite (Fig. 4, Table 4), this seems to be in contradiction to your soil profile data (Table 5). Please elaborate on this.

*We have a slightly unusual situation in that catchment in addition to the major drought during the study period. The low impact of interceptions storage evaporation on the isotopic composition of throughfall seems counterintuitive. We think that the precipitation characteristics reduce this effect at this forested site. Summer precipitation (with highest expected impacts) usually occurs as a few convective (higher intensity) events. These characteristics lead to little dripping and complete emptying of interception storages until the next precipitation event (which hardly occurred during the study period anyway). Enriched signals in forest soils are therefore linked to the process of soil evaporation fractionation.*
*Nevertheless, we cannot exclude canopy effects for other stands in the catchment or even generally for this site as we did not sample stemflow.*

L. 223

Please delete one of the "in" before "Figure 4"

*Done*

L. 225

Is the sample number of precipitation really higher than throughfall? According to Table

precipitation (68), throughfall (136). Please clarify.

*You are right. we changed that.*

L. 232

I like the heat map (Figure 5), but you could think about providing a figure for each site with the soil profile isotope data as supplemental material.

*The heatmap enables easy visual interpretation of the isotopic dynamics. (But we will add soil profiles as supplemental material).*

L. 238-240

Not clear what you mean, please rephrase.

*Rephrased*

L. 259

'upper' instead of "top"

*Done as suggested.*

L. 260

Please insert 'the upper' before "three of the. . ."

*Done.*

L. 289

Insight 'into' instead of "in"

*Done.*

L. 316

Please replace "soils" by 'soil layers'.

*Thank you. We adjusted it.*

L. 324
Stick to "α-values" to be consistent with L. 323 and Table 6.

*We changed it.*

L. 338

Please insert 'in' or 'reported from' before ". . .the previous winter".

*We added 'in'.*

L. 342

'storage,' instead of "stores"
*Thank you. We changed it as recommended.*
L. 356-357

". . .urgency with the by. . ." ??? Please rephrase.

*Corrected.*

L. 369

". . .headwater. . ." is 'catchment' missing?

*Added 'catchment'.*

L. 376

Please add 'the two' between "between" and "sites"

*Done.*

L. 378

Please insert '1' before "year old)"
*Corrected.*

Fig. 1

There is enough space to put the overview beneath the detailed maps – makes the layout of the figure a bit clearer.

Right part: please replace "Landuse" by 'Soils'

*Legend title was adapted.*

Fig. 3

Legend: 'h' is missing in "Troughfall"

*Corrected.*

Fig. 5

Please label forest and grassland.

*Corrected*

Table 4

Deltas are missing in header

Are 5th and 95th percentile of d18-O identical for precipitation and throughfall? Stream:
5th and 95th percentile of d18-O both -8.6? Please double-check.

*We double-checked table 4 and precipitation and throughfall numbers are correct. Stream 95th percentile was corrected. Thank you!*

Table 5

Caption: 'soil water isotope samples' instead of "soil isotopes samples"

*Thank you. We changed 'soil isotope samples' to 'bulk soil water isotope samples' to clarify.*

Global changes

I would prefer "and" instead of "&" (e.g. l. 62, 112, 136)

*We changed many '&' to 'and'.*

please change "stable water isotopes" to 'water stable isotopes' check '-' vs. '–' throughout the manuscript (e.g. L. 287, L. 291)

*We checked and corrected this throughout the manuscript.*

---

## Author Response (AR1)

*Throughout the revision of the manuscript, the authors have corrected the comments by the two anonymous reviewers and the editor and have addressed them accordingly. All comments (black) and responses (blue) are listed below and line numbers in brackets indicate the position of adaptations in the track changed manuscript in this document. Removed text in the manuscript is indicated as red and crossed out and added text as blue and underlined.*

**Anonymous Referee #1**

General comments

In the manuscript entitled 'Using isotopes to understand evaporation, moisture stress and re-wetting in catchment forest and grassland soils of the summer drought of 2018' the question is addressed, how the drought 2018 affected two different land-use/soil community sites in a catchment in NE Germany. The two plot sites were monitored during a period of water stress, when the catchment could no longer sustain blue water fluxes (e.g. stream flow) or green water needs (e.g. crop production), and the subsequent recovery.

Ecosystem response to this climatic anomaly is investigated by using water stable iso-tope data of precipitation and throughfall, stream-, groundwater and especially from soil water profiles. Monthly soil profile samples in 6 different depths down to 1 m under two different land-use types were taken from September 2018 to February 2019. Soil water isotopes were analysed using direct vapor equilibration laser spectrometry (DVE-LS). These data were used to estimate mean transit times (MTT) in the soils at the different depths as well as young water fractions, using a fitted sine-wave method. Based on collected meteorological and sap flow data ET-pot was calculated. Soil moisture was monitored in three different depths at both sites. Drought severity was quantified with the SPI, based on long-term precipitation data from the DWD.

It could be shown that the forest soils were dominated by rapid young water fluxes after rainfall events whereas the grassland soils were more retentive and dominated by older water. It is concluded that implications for blue and green water management should be investigated in a greater range of representative vegetation/ soil units and that further research efforts on climate change and management adaptations in the critical zone of drought sensitive ecosystems is needed.

Overall, the manuscript is well structured and nicely written.

The topic fits well to the scope of the journal and appears to be of interest for the readers; I only suggest moderate revisions prior to acceptance and publication in Hydrology and Earth System Sciences.

*We thank the reviewer 1 for the encouraging comments on our manuscript. We are grateful for this very detailed and careful review of our work.*

**Specific comments**

Title

Please add which isotopes were investigated, either 'water stable isotopes' or 'stable isotopes (d18-O, d2-H)'.

*We specified this as 'water stable isotopes' in the manuscripts title. (Title)*

Evaporation or Evapotranspiration?

*We choose evaporation as the focus of the isotope techniques is on evaporation. (Title)*

L. 4
Please add 'water' before ". . .stable isotopes to. . ."

*'water' was added to particularize the isotopes used. (L.4)*

L. 47
Compared to e.g. soil moisture probes, laser absorption spectroscopy is not really "inexpensive". . . I wouldn't go too deep into the history of stable isotope measurement techniques, but it could be added that compared to traditional mass spec. techniques, laser absorption spectroscopy is relatively inexpensive. Mentioning that the invention of laser absorption spectroscopy has facilitated several new techniques in the last ten years would also emphasise that your approach is relatively new.

*Thanks for the suggestion. We have added more detail on that. (L. 48-51)*

L. 52
Developed by whom? Not by Hendry et al., maybe they improved. Please delete "e.g.," in the brackets.

*Thank you. This was indeed very misleading. We adapted the text. (L. 54-55)*

L. 60
Shift "To" to line 59 after "study"

*Thank you. We adjusted it. (L. 62)*

L. 66
'located' instead of "based"

*Changed according to suggestion. (L.69)*

L. 123
Throughfall was sampled as well at 1 m height? Which distance between the five gauges?

*We added more detail regarding the experimental setup of the throughfall to the revised manuscript. (L. 126-128)*

L. 131
blank is missing between "for" and "d2H"

*Adjusted. Thank you. ( L. 142)*

L. 132
Please complete ". . .from October 2018 to. . ."

*We completed this (to February 2019; L. 143)*

L. 143
Please provide part-no. of sample bags

*Thank you. Indeed part-no changed in comparison to older studies. (L. 155)*

L. 146
Please insert 'gas' after "headspace"

*Done as suggested. (L. 157)*

L. 147
What kind of 'standards' were used?

*We added more detail to the standards (liquid/ 10ml) but too much detail might be confusing as these standards are only used for calibration and not relevant for the further storyline. (L. 158-159)*

L. 150
d-Excess should be introduced somewhere here.

*We included a short introduction to the d-excess concept. (L. 172-175)*

L. 150-152
Method or results section?
*We presented this information here to give insight into the method and its precision. (L. 162-164)*

L. 157
Please add 'oil' after "paraffin"

*Thank you. We specified it to "paraffin oil". (L. 136)*

L. 171
Where can I see "young water" in figure 2?
*This was a mistake – we removed the reference to Fig 2 here. (L. 188)*

L. 171-172
Method or results section?

*We mentioned this here to clarify the definition of young water in the applied method. (L. 188-189)*

L. 213
One would always expect slightly evaporated signals in throughfall (enriched in heavy isotopes) compared to precipitation. In your study it is opposite (Fig. 4, Table 4), this seems to be in contradiction to your soil profile data (Table 5). Please elaborate on this.

*We have a slightly unusual situation in that catchment in addition to the major drought during the study period. The low impact of interceptions storage evaporation on the isotopic composition of throughfall seems counterintuitive. We think that the precipitation characteristics reduce this effect at this forested site. Summer precipitation (with highest expected impacts) usually occurs as a few convective (higher intensity) events. These characteristics lead to little dripping and complete emptying of interception storages until the next precipitation event (which hardly occurred during the study period anyway). Enriched signals in forest soils are therefore linked to the process of soil evaporation fractionation.*
*Nevertheless, we cannot exclude canopy effects for other stands in the catchment or even generally for this site as we did not sample stemflow. (L.233-235)*

L. 223
Please delete one of the "in" before "Figure 4"

*Done. (L.242)*

L. 225
Is the sample number of precipitation really higher than throughfall? According to Table precipitation (68), throughfall (136). Please clarify.

*You are right. we changed that. (L. 244)*

L. 232
I like the heat map (Figure 5), but you could think about providing a figure for each site with the soil profile isotope data as supplemental material.

*The heatmap enables easy visual interpretation of the isotopic dynamics. (But we will add soil profiles as supplemental material).*

L. 238-240
Not clear what you mean, please rephrase.

*Rephrased. (L258-259)*

L. 259
'upper' instead of "top"

*Done as suggested. (L. 278)*

L. 260
Please insert 'the upper' before "three of the. . ."

*Done. (L. 279)*

L. 289
Insight 'into' instead of "in"

*Done. (L. 311)*

L. 316
Please replace "soils" by 'soil layers'.

*Thank you. We adjusted it. (L. 340)*

L. 324
Stick to "α-values" to be consistent with L. 323 and Table 6.

*We changed it. (L. 348)*

L. 338
Please insert 'in' or 'reported from' before ". . .the previous winter".

*We added 'in'. (L. 362)*

L. 342

'storage,' instead of "stores"
*Thank you. We changed it as recommended. (L. 367)*

L. 356-357
". . .urgency with the by. . ." ??? Please rephrase.

*Corrected. (L. 381-382)*

L. 369
". . .headwater. . ." is 'catchment' missing?

*Added 'catchment'. (L. 394)*

L. 376
Please add 'the two' between "between" and "sites"

*Done. (L. 405)*

L. 378
Please insert '1' before "year old)"
*Corrected. (L. 407)*

Fig. 1
There is enough space to put the overview beneath the detailed maps – makes the layout of the figure a bit clearer.

Right part: please replace "Landuse" by 'Soils'

*Legend title was adapted.*

Fig. 3
Legend: 'h' is missing in "Troughfall"

*Corrected.*

Fig. 5
Please label forest and grassland.

*Corrected*

Table 4
Deltas are missing in header

Are 5th and 95th percentile of d18-O identical for precipitation and throughfall? Stream: 5th and 95th percentile of d18-O both -8.6? Please double-check.

*We double-checked table 4 and precipitation and throughfall numbers are correct. Stream 95th percentile was corrected. Thank you!*

Table 5
Caption: 'soil water isotope samples' instead of "soil isotopes samples"

*Thank you. We changed 'soil isotope samples' to 'bulk soil water isotope samples' to clarify.*

**Global changes**

I would prefer "and" instead of "&" (e.g. l. 62, 112, 136)

*We changed many '&' to 'and'. (e.g. L. 65,115, 147, 148, 241)*

please change "stable water isotopes" to 'water stable isotopes' check '-' vs. '–' throughout the manuscript (e.g. L. 287, L. 291)

*We checked and corrected this throughout the manuscript. (e.g. L. 389)*

**Anonymous Referee #2**

The manuscript entitled "Using isotopes to understand evaporation, moisture stress and re-wetting in catchment forest and grassland soils of the summer drought of 2018." by Lukas Kleine, Dörthe Tetzlaff, Aaron Smith, Hailong Wang, and Chris Soulsby presents an interesting contribution to our understanding of ecohydrological processes in a mixed land cover catchment (forest and agricultural), especially under the influence of climate anomalies. The authors conducted a case study in North-East Germany in the Demnitzer Millcreek catchment. They highlight the use of isotopic tracers together with conventional hydrology to understand the effect of drought progress, the recovery of soil water storage and the memory effect of drought evaporation when the catchment could no longer hold streamflow and crop production and further mixing with fresh precipitation.

The study shows an important work with a logical structure and is clearly written, in my opinion, it deserves to be considered for publication in the HESS after some minimal revisions. Most of my editing comments match those of Referee 1 and have already been addressed by the authors.

*We are grateful for the constructive comments of reviewer 2 on the manuscript. We appreciate this thoughtful and stimulating review of our work. Throughout the revision of the manuscript, the authors have adapted the terminology used to improve clarity for the discussion and key messages. For the specific comments, we have addressed them accordingly. Responses are given below and will be incorporated in the revised manuscript.*

I recommend the authors to be careful when using the terms "blue and green water", as it is broad and varied in the literature, so I suggest that they highlight in the introduction section what they specifically refer in this study.
*We clarified the use of terms "blue" (as groundwater recharge and stream discharge) and "green" (evapotranspiration) water fluxes in the introduction section. (L. 7-8). We would like – respectfully – keep these terms as they are important and widely used terms in the literature.*

I'm a little concerned about the limited availability of soil water isotope samples (monthly basis) used to drive such a conclusion based on tentative MTTs. The manuscript would benefit for a wider discussion and to clearly state this limitation. In order to reaffirm the credibility of these results, I suggest widening the context of the study by comparing it with similar drought cases in nearby sites or with comparable geographical regions.

*We agree, and recognise of the limitations of monthly destructive samples and we are careful to be circumspect about the inferences. Still, as the work getting such samples is so labour intensive, not many such data sets exist. To assess spatial variability (replicates) and enable ongoing sampling in the limited site area beyond the study period, we had to limit the temporal resolution of the sampling. Nevertheless, the insight in subsurface processes was invaluable and demonstrated the efficiency of this method for a*

*first approximation. We will expand the critical discussion but cannot widen the context of the actual study (obviously) as this would be a different paper. To our knowledge, no other bulk water isotope samples exist from nearby sites (or at least are not published yet). Though we make comparison to chloride related water ages from lysimeters for another site in Brandenburg. (L. 295-296, 356-358)*

Further, an extended amount of literature pointed out that MTT (based is a gamma distribution with two parameters and derived MTTs concept) is only a qualitative indicator of catchments for a first screen and basic comparison, however a bit critical when the evolution of water ages is involved. With the available information, I firmly believe that it would be possible to obtain better and accurate results by including more elaborate and non-stationary criteria in the analysis.

*We agree with your concern and are well aware of the limitations of this method. We tried to emphasise the tentative nature of these results. Further this concern lead to the additional consideration of Young water fractions as an unbiased indictor of water ages. This largely supported the MTT results and helped underpin our conclusions. The basic nature of this analysis is further highlighted in the manuscript (L. 178, 403)*
*Also, we added a reference to a process-based ecohydrological modelling approach (considering isotopes) at these two plot sites which also estimated the water ages in the conclusion (recently published by Smith et al.; L. 412).This is also broadly consistent with the more basic approximations reported here. However, conducting such complex non-stationary analysis would be beyond the scope of the paper.*

Finally, please improve figure 4, the size of the symbols and the colours used make it difficult to identify isotopic signatures.

*We adapted color-codes and sizes to make it easier to identify.*

**Editor comments**
As you can see the two reviewers were rather positive on your study. The comments they raised are well addressed by your reply. So I invite you to make these changes accordingly.

*Thank you. We will include the changes in the manuscript.*

Additionally, I have some minor comment too:
- L187-188: unit should be mm/d

*Thank you. We changed it from mm to mm/d. (L. 204-205)*

- L212: an interception percentage of 7% is rather low, so likely the canopy is quite open, resulting in a significant amount of throughfall that stems from direct rainfall. This might explain the little enrichment in throughfall water. It would also be nice to refer to https://doi.org/10.1002/wat2.1187, where a nice overview is given on the effect of interception on throughfall.

*Thank you! We added the reference and elaborated on this. I double checked numbers and added more details.( L. 229 & 317-319)*

- Fig 4: this is the forest site?

*This is daily precipitation from the automatic weather station and throughfall from the forested site. We altered the Figures caption to make it clearer.*

- Fig 5: Please work with sub a) and b) to clarify the left and right figure as forest and grass. Right?

*Reviewer 1 had a similar comment and we added site names on top.*

[revised manuscript text omitted]